# Bleeding complications of thromboprophylaxis with dabigatran, nadroparin or rivaroxaban for 6 weeks after total knee arthroplasty surgery: a randomised pilot study

Lucia van der Veen,[1,2] Marijn Segers,[1] Jos JAM van Raay,[3] Carina LE Gerritsma-Bleeker,[3] Reinoud W Brouwer,[3] Nic JGM Veeger,[4] Marinus van Hulst [1,5]

**To cite:** van der Veen L, Segers M, van Raay JJAM, et al. Bleeding complications of thromboprophylaxis with dabigatran, nadroparin or rivaroxaban for 6 weeks after total knee arthroplasty surgery: a randomised pilot study. BMJ Open 2021;11:e040336. doi:10.1136/bmjopen-2020-040336

For numbered affiliations see end of article.

**Correspondence to**
Dr Marinus van Hulst;
m.van.hulst@rug.nl

## ABSTRACT

**Objectives** For the non-vitamin-K oral anticoagulants, data on bleeding when used for 42 days as thromboprophylaxis after total knee arthroplasty (TKA) are scarce. This pilot study assessed feasibility of a multicentre randomised clinical trial to evaluate major and clinically relevant non-major bleeding during 42-day use of dabigatran, nadroparin and rivaroxaban after TKA.

**Patients and methods** In 70 weeks, between July 2012 and November 2013, 198 TKA patients were screened for eligibility in the Martini Hospital (Groningen, the Netherlands). Patients were randomly assigned to dabigatran (n=45), nadroparin (n=45) or rivaroxaban (n=48). The primary outcome was the combined endpoint of major bleeding and clinically relevant non-major bleeding. Secondary endpoints of this study were the occurrence of clinical venous thromboembolism (VTE) (pulmonary embolism or deep venous thrombosis), compliance, duration of hospital stay, rehospitalisation, adverse events and Knee Injury and Osteoarthritis Outcome Score (KOOS).

**Results** The primary outcome was observed in 33.3% (95% CI 20.0% to 49.0%), 24.4% (95% CI 12.9% to 39.5%) and 27.1% (95% CI 15.3% to 41.8%) of patients who received dabigatran, nadroparin or rivaroxaban, respectively (p=0.67). Major bleeding was found in two patients who received nadroparin (p=0.21). Clinically relevant non-major bleeding was observed in 33.3% (95% CI 20.0% to 49.0%), 22.2% (95% CI 11.2% to 37.1%) and 27.1% (95% CI 15.3% to 41.8%) for dabigatran, nadroparin and rivaroxaban, respectively (p=0.51). Wound haematoma was the most observed bleeding event. VTE was found in one patient who received dabigatran (p=0.65). The presurgery and postsurgery KOOS qQuestionnaires were available for 32 (71%), 35 (77%) and 35 (73%) patients for dabigatran, nadroparin and rivaroxaban, respectively. KOOS was highly variable, and no significant difference between treatment groups in mean improvement was observed.

**Conclusions** A multicentre clinical trial may be feasible. However, investments will be substantial. No differences in major and clinically relevant non-major bleeding

### Strengths and limitations of this study

► In this study, dabigatran, rivaroxaban and nadroparin were directly compared as thromboprophylaxis after total knee arthroplasty surgery.
► We evaluated extended thromboprophylaxis for 42 days after total knee arthroplasty surgery.
► As this is a pilot study, hence, limited number of patients was evaluated.

events were found between dabigatran, nadroparin and rivaroxaban during 42 days after TKA. KOOS may not be suitable to detect functional loss due to bleeding.

**Trial registration number** NCT01431456.

## INTRODUCTION

After total knee arthroplasty (TKA) surgery, patients are at risk to develop venous thromboembolism (VTE). Without thromboprophylaxis, (venographic) deep venous thrombosis (DVT) can be found in 40%–80% of patients.[1 2] The risk of non-fatal and fatal pulmonary embolism (PE) for TKA surgery patients without thromboprophylaxis is 1.8%–7.0% and 0.2%–0.7%, respectively.[3] Hence, thromboprophylaxis is indicated for all patients undergoing TKA. Low-molecular-weight heparins (LMWHs) have become a gold standard in preventing VTE. Unfortunately, these anticoagulants have disadvantages such as a subcutaneous route of administration and risk of developing heparin-induced thrombocytopenia. These disadvantages can lead to low adherence of orthopaedic surgeons to the thromboprophylaxis guidelines and hamper compliance by patients.[4] During the last decades, oral anticoagulants were developed. Dabigatran,

rivaroxaban, apixaban and edoxaban are four available non-vitamin-K oral anticoagulants (NOAC).

In clinical trials as well as in a real-world data study, rivaroxaban was superior in preventing VTE compared with enoxaparin in TKA surgery patients.[5–7] However, compared with enoxaparin, rivaroxaban was associated with a slightly increased risk of postoperative bleeding and wound complications.[8] Clinical trials and subsequent meta-analyses showed that VTE risk and bleeding risk are similar for dabigatran compared with LMWH in TKA.[8–11]

The external validity of the rivaroxaban and dabigatran clinical trials may be limited for Dutch TKA patients. Next to that, enoxaparin was used in the clinical trials, whereas in the Netherlands, nadroparin is predominantly used; also, the duration is extended in the Netherlands.[12] In the market authorisation for dabigatran and rivaroxaban, the duration of VTE prevention is limited to 10–14 days. In our study, thromboprophylaxis was offered to patients for a period of 42 days after surgery in agreement with Dutch and regional guidelines.[12 13] Moreover, clinical trials have demonstrated that a more prolonged prophylaxis with LMWH after hospital discharge significantly reduces the incidence of venographically detected DVT and PE after arthroplasty.[14 15]

The objective of our study was to compare the 42-day use of dabigatran, nadroparin and rivaroxaban on safety after TKA in a randomised open-label pilot study by assessing the risk of major and clinically relevant non-major bleeding using a standardised model of bleeding definitions,[16] in order to obtain insight into the design of a multicentre study investigating the safety of the new oral anticoagulants for prevention of VTE after TKA in the Dutch setting. The rationale of our study was previously published.[17]

## METHODS
### Study design
This study was an open-label randomised pilot trial with a three-arm design, conducted at the Martini Hospital in Groningen, the Netherlands. This clinical trial was registered at ClinicalTrials.gov and opened for enrolment in July 2012. At the start of the study, an independent hospital pharmacist assigned the three treatment groups to a consecutive series of numbers, using a computer-generated randomised list. Before surgery, patients were included and randomly assigned to one of the three treatment groups (dabigatran, rivaroxaban or nadroparin). The allocation concealment was reached by using the sequentially numbered opaque sealed envelope method.[18] On enrolment, each patient was assigned to the next consecutive treatment number, and the corresponding study medication was dispensed in an open-label fashion.

### Patient and public involvement
Patients or the public were not involved in the design or reporting or dissemination plans of our research as this

study is a pilot study. Patients were involved in the conduct of the trial by sharing experiences during the visits.

### Study population
The target population consisted of TKA patients. Patients were recruited from the outpatient orthopaedic surgery clinic at the Martini Hospital in Groningen by the orthopaedic surgeon or trainee. Patients ≥18 years old and weighing more than 40 kg who were scheduled for primary elective TKA and had provided signed informed consent were eligible for the study. Exclusion criteria included a known inherited or acquired clinically significant active high risk of bleeding or bleeding disorder; major surgery, trauma, uncontrolled severe arterial hypertension or myocardial infarction within the last 3 months; history of acute intracranial disease or haemorrhagic stroke; gastrointestinal or urogenital bleeding or ulcer disease within the last 6 months; cirrhotic patients with moderate hepatic impairment (aspartate or alanine aminotransferase levels higher than twice the upper limit of the normal range within the last month); severe renal insufficiency (creatinine clearance <30 mL/min); other indication for treatment with anticoagulants; active malignant disease; pregnancy or breastfeeding. Patients received standard orthopaedic care and physiotherapy according to the local standardised protocol. During the enrolment of the trial, fast-track knee replacement surgery was implemented.[19 20]

### Treatment regimens
Patients were assigned to oral dabigatran etexilate 150 mg or 220 mg (two capsules, 75 mg or 110 mg) once daily, subcutaneous nadroparin 2850 IU =0.3 mL once daily or 10 mg of oral rivaroxaban once daily. The first dose of dabigatran was one-half of subsequent doses (one capsule, 75 mg or 110 mg). Treatment was continued for a total of 42 days after TKA.

### Outcome measures
The primary endpoint of this study was a major bleeding event and/or clinically relevant non-major bleeding event within the 42-day treatment period. Any bleeding event that occurred during the 42-day treatment period, perceived by patient, researcher, nurse, orthopaedic surgeon or other health worker, was registered and evaluated. A blinded independent expert adjudication committee classified all bleeding events according to international guidelines.[16]

A major bleeding was defined as fatal bleeding, clinically overt bleeding associated with a decrease in the haemoglobin level of more than 20 g/L compared with the prerandomisation level, clinically overt bleeding leading to transfusion of ≥2 units of whole blood or packed cells, critical bleeding (intracerebral, intraocular, intraspinal, pericardial or retroperitoneal), bleeding warranting treatment cessation and bleeding located at the surgical site and leading to reoperation or to any unusual medical intervention or procedure

for relief (eg, draining or puncture of a haematoma at the surgical site and transfer to an ICU or emergency room).

Clinically relevant non-major bleeding was defined as spontaneous skin haematoma >25 cm$^2$; wound haematoma >100 cm$^2$; spontaneous nose bleeding or gingival bleeding lasting longer than 5 min; spontaneous rectal bleeding creating more than a spot on toilet paper; macroscopic haematuria either spontaneous or, if associated with an intervention, lasting longer than 24 hours; and other bleeding events considered clinically relevant by the investigator not qualifying as a major bleeding.

Secondary endpoints of this study were the occurrence of clinical VTE (PE or DVT), compliance, duration of hospital stay, rehospitalisation, adverse events and Knee Injury and Osteoarthritis Outcome Score (KOOS).[21]

The duration of hospital stay and rehospitalisation were retrieved from electronic medical records.

Compliance was measured by counting the number of doses left over after 42 days of treatment. In our study, patients were deemed compliant if patients didn't miss a single dose. Adverse events were included in the analysis if the relationship with the study drug was possible, probable or highly probable.

The KOOS (www.koos.nu and online supplemental file 1) is a 42-item self-administered questionnaire that includes five dimensions: pain, disease-related symptoms, activities of daily living function (ADL), sport and recreation function and knee-related quality of life measured using a Likert Scale (0–4 scale).[21] The KOOS was administered presurgery and 6 weeks after TKA. A patient with an improvement of ten points or more on a KOOS dimension after TKA was defined as responder.

### Statistical analysis

The absolute bleeding risk was estimated as the percentage of patients with a bleed for each treatment group, with the associated 95% confidence intervals. Confidence intervals were derived using the (exact) Clopper-Pearson method. The risk of thromboembolic events was assessed using the same methodology. The analysis was based on the per-protocol population, that is, all fully compliant patients that were evaluated 42 days after surgery. One-way analysis of variance was used for analyses of continuous variables between multiple groups. If normality assumption was violated (Shapiro-Wilk test for normality), the non-parametric Kruskal-Wallis test was used. Distribution of nominal variables for different groups was tested for statistical significance using the exact $\chi^2$ test. All statistical analyses were performed using IBM Statistics SPSS V.25.0.0. Due to the explorative nature of this pilot study, all p values will be interpreted as explorative rather than confirmative.[22] The expected sample size for the design of a two-arm clinical trial based on our findings was calculated using the SampleSize4ClinicalTrials package in R (R Studio V.1.3.959/R V.4.0.3).

## RESULTS

Between July 2012 and November 2013, a total of 198 patients were assessed for eligibility of which 148 patients were randomised to treatment. In total, 138 patients were included in the final analysis; see flow diagram (figure 1). The three treatment groups were well balanced in terms of baseline demographics (table 1). Of all patients, 63% were woman, with a mean age of 66 years and a mean weight of 89 kg. The mean time from end of surgery to the first study drug intake was 8 hours and 48 min. In total, 111 (80%) patients had fast-track TKA. Compared with patients in the other treatment groups, a larger share of patients in the dabigatran group had fast-track TKA. Thirty-three (24%) patients received a patella prosthesis during TKA. Compared with the nadroparin group, a larger share of patients in the rivaroxaban group and smaller share of patients in the dabigatran group received a new patella. Presurgery KOOS was well balanced between the three treatment groups.

### Primary endpoints

The combined endpoint of major and clinically relevant non-major bleeding was observed in 15 of 45 (33.3%; 95% CI 20.0% to 49.0%) patients who received dabigatran, 11 of 45 (24.4%; 95% CI 12.9% to 39.5%) patients who received nadroparin and 13 of 48 (27.1%; 95% CI 15.3% to 41.8%) patients who received rivaroxaban. There was no significant difference in these bleeding events between dabigatran, rivaroxaban and nadroparin (p=0.67); see table 2.

Major bleeding events occurred in none of 45 (0%; 95% CI 0% to 7.9%) patients in the dabigatran group, 2 of 45 (4.4%; 95% CI 0.5% to 15.1%) patients in the nadroparin group and none of 48 (0%; 95% CI 0% to 7.4%) patients in the rivaroxaban group. One of these events was a clinically overt bleeding from the wound leading to transfusion of two or more units of whole blood or packed cells, and one was adjudicated as a subdural haematoma caused by a fall. There was no difference in major bleeding events between dabigatran, nadroparin and rivaroxaban.

Clinically relevant non-major bleeding events were observed in 15 patients who received dabigatran (33.3%; 95% CI 20.0% to 49.0%), 10 patients who received nadroparine (22.2%; 95% CI 11.2% to 37.1%) and 13 patients who received rivaroxaban (27.1%; 95% CI 15.3% to 41.8%). Most of these clinically relevant non-major bleeding events were wound haematomas. There was no difference in clinically relevant non-major bleeding events between dabigatran, rivaroxaban and nadroparin. The last major or clinically relevant non-major bleeding event was observed a median 7, 7 and 6 days postoperatively, for dabigatran, nadroparin and rivaroxaban, respectively. In the period between 14 and 42 days after surgery, one major or clinically relevant non-major bleeding event was observed in all treatment groups. The overall risk of major and clinically relevant non-major bleeding was 39 events in a total intention to treat population of 148

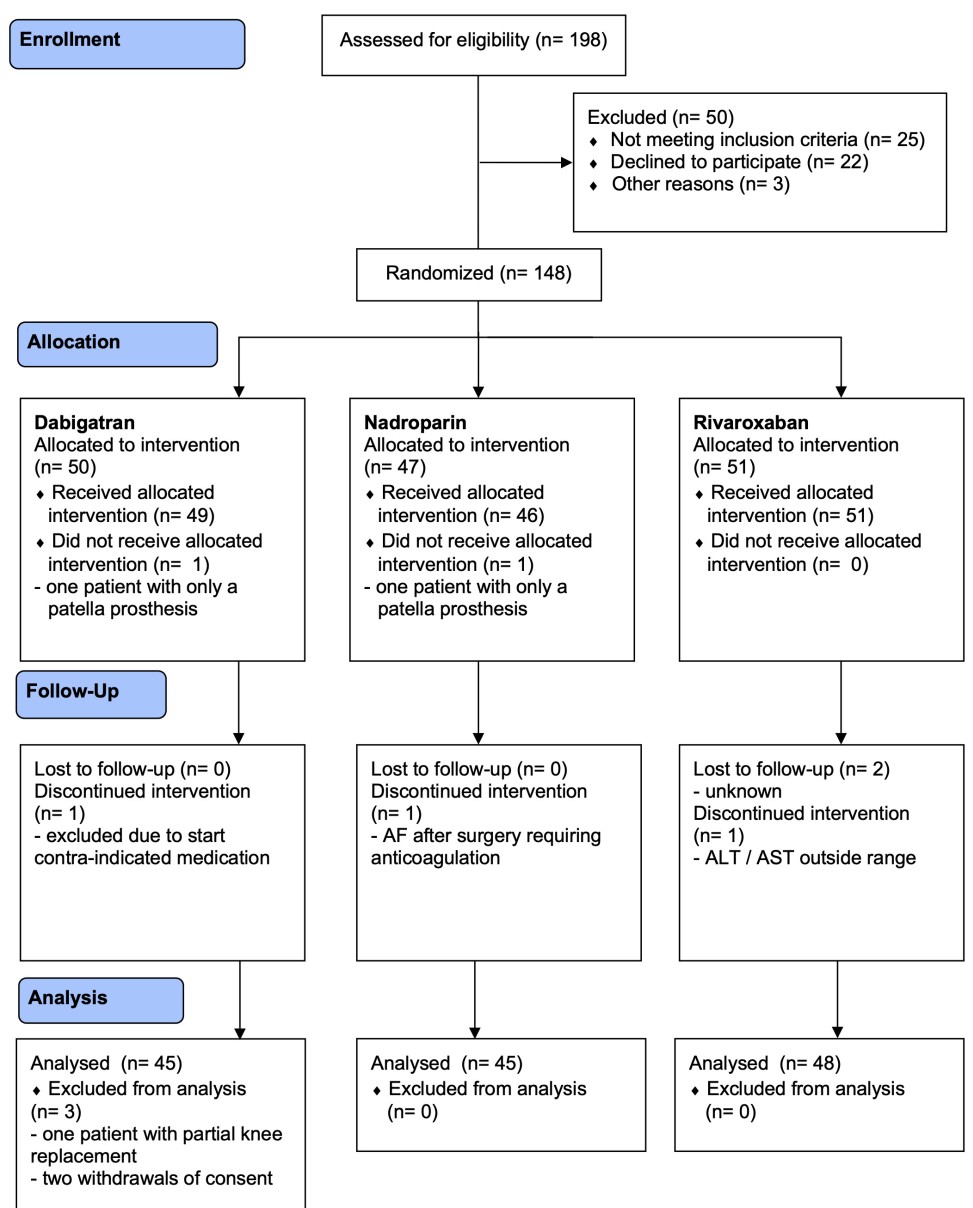

**Figure 1** Flow diagram. AF, atrial fibrillation; ALT, alanine aminotransferase; AST, aspartate aminotransferase.

randomised patients, that is, a proportion of 0.26. Using a relative risk difference of 10%, that is, an absolute difference of 2.6%, we estimated that 4468 patients per treatment group would be needed in a two-arm non-inferiority design with a statistical power of 80% and a two-sided type I error rate of 2.5%.

### Secondary endpoints

The secondary endpoints of this study were occurrence of clinical VTE, hospital stay, readmission, adverse events, compliance and KOOS; see tables 2 and 3. Clinical VTE occurred in one of 45 patients receiving dabigatran and none in those receiving rivaroxaban or nadroparin. Unfortunately, this patient with VTE developed PE and died. The median duration of hospital stay was 3, 3 and 4 days for dabigatran, nadroparine and rivaroxaban, respectively. Readmission within 3 months postsurgery was not different between treatment groups. The number

of patients readmitted within 3 months was two (4.4%), four, (8.9%) and three (6.3%) patients, for dabigatran, nadroparin and rivaroxaban, respectively. Adverse events were reported in nine (20%), two (4.4%) and five (10.4%) patients for dabigatran, nadroparin and rivaroxaban, respectively. Five patients in the dabigatran group (11.1%) experienced gastrointestinal adverse effects such as abdominal pain, diarrhoea and/or indigestion. In the rivaroxaban group, gastrointestinal adverse effects were found in two patients (4.4%).

In the dabigatran group, 27 of 44 patients (61.4%) were compliant versus 25 of 44 patients (56.8%) in the nadroparine group and 25 of 48 patients in the rivaroxaban group (52.1%). Compliance was not different between treatment groups. Also, the number of missed days of treatment was not different between treatment groups. In the period between the 42-day follow-up and

van der Veen L, *et al. BMJ Open* 2021;**11**:e040336. doi:10.1136/bmjopen-2020-040336

**Table 1** Patient characteristics

| | Dabigatran (n=45) | Nadroparine (n=45) | Rivaroxaban (n=48) | Total (n=138) |
|---|---|---|---|---|
| Female, n (%) | 26 (58) | 31 (69) | 30 (63) | 87 (63) |
| Age (year) (±SD) | 66±7.9 | 67±10 | 65±7.5 | 66±8.7 |
| Weight (kg) (±SD) | 88±12 | 89±18 | 90±21 | 89±17 |
| Fast-track surgery, n (%) | 41 (91) | 35 (78) | 35 (73) | 111 (80) |
| Received new patella, n (%) | 7 (16) | 10 (22) | 16 (33) | 33 (24) |
| History of VTE, n (%) | 1 (2.2) | 1 (2.2) | 2 (4.2) | 4 (2.9) |
| Time to first dose (hours) (±SD) | 8.6±2.3 | 9±2.2 | 8.8±2.1 | 8.8±2.2 |
| Presurgery KOOS (±SD) | | | | |
| Pain (n) | 47±15 (32) | 48±20 (35) | 51±21 (35) | 49±19 (102) |
| Symptoms (n) | 55±16 (32) | 49±18 (35) | 50±18 (35) | 51±17 (102) |
| ADL (n) | 52±15 (32) | 48±19 (35) | 56±19 (35) | 52±18 (102) |
| Sport and recreation (n) | 23±25 (29) | 24±28 (31) | 22±22 (29) | 23±25 (89) |
| Quality of life (n) | 30±12 (31) | 30±16 (35) | 30±15 (35) | 30±14 (101) |

ADL, activities of daily living function; KOOS, Knee Injury and Osteoarthritis Outcome Score; VTE, venous thromboembolism.

3 months postsurgery, we observed no additional bleeding and VTE events.

The presurgery and postsurgery KOOS Questionnaires were available for 32 (71%), 35 (77%) and 35 (73%) patients for dabigatran, nadroparin and rivaroxaban, respectively (table 3). No difference between treatment groups in mean improvement was observed for the KOOS dimensions pain, symptoms, ADL, sport and recreation and quality of life. Also, the proportion of responders, patients with a ten point or more improvement on a KOOS dimension, was not different between dabigatran, nadroparin and rivaroxaban. The SD in KOOS was mostly equal or even higher than the mean.

## DISCUSSION

After knee replacement, when bleeding in the replaced joint can have serious consequences, the balance between inducing bleeding and preventing VTE is pivotal in thromboprophylaxis. In this study, we compared bleeding complications of dabigatran, rivaroxaban and nadroparin when used to prevent VTE after TKA surgery. This exploratory study shows that there was no significant difference in major and clinically relevant bleeding between dabigatran, rivaroxaban and nadroparin. The bleeding risk we found is higher than observed in previous knee replacement studies. In such studies, a risk of non-major bleeding events of between 1% and 6.8% was observed.[5 6 23–25] In our study, the risk of bleeding events is higher mainly due to the risk of wound haematomas >100 cm². A possible explanation for the difference in occurrence of clinically relevant non-major bleeding events is our strict adherence to EMA bleeding definitions.[16 24] For instance, in the RECORD 3 and RECORD 4, rivaroxaban trials, surgical site and wound haematomas were not reported.[5 6 8 24] In the EPCAT II trial, extended VTE prophylaxis comparing rivaroxaban to aspirin following total hip and knee arthroplasty, only infected wound haematomas were considered clinically relevant.[25]

The market authorisation for the prevention of VTE after TKA for dabigatran and rivaroxaban is 10–14 days. In our study, patients received thromboprophylaxis for 42 days. Most bleeding events occurred during the first 14 days postsurgery. Notably, bleeding risks after the first 14-day use of dabigatran, nadroparin and rivaroxaban were low and not different between treatment groups. Based on our pilot study, a meta-analysis of dabigatran and rivaroxaban in TKA and the EPCAT II trial, it is expected that large numbers of patients have to be included in a clinical trial to detect a clinically relevant difference in major and clinically relevant non-major bleeding between NOACs and LMWHs when used for 42 days.[8 25] Therefore, costs of conducting a multicentre clinical trial based on the evaluated protocol and a formal power calculation are expected to be very high.

As expected, the number of symptomatic VTE cases was very low. Unfortunately, one case of PE occurred in the dabigatran group, leading to the death of the patient.

It is generally assumed that patients would prefer the oral route to subcutaneous injections. However, it is unknown if the route of administration has an impact on patient compliance. In our pilot study, we found no clear indication for a difference in compliance between rivaroxaban, nadroparin and dabigatran.

Although we found no differences in the number of patients with adverse reactions, the number of patients with gastrointestinal adverse reaction in the dabigatran group appeared to be higher than in the other treatment groups. Dabigatran is also in other clinical trials associated with gastrointestinal adverse effects.[26] Between dabigatran, nadroparin and rivaroxaban, no difference in

**Table 2** Primary and secondary outcomes of thromboprophylaxis with dabigatran, nadroparin or rivaroxaban after total knee arthroplasty

| | Dabigatran (n=45) | | Nadroparin (n=45) | | Rivaroxaban (n=48) | | P value |
|---|---|---|---|---|---|---|---|
| | n | % | n | % | n | % | |
| **Primary outcome** | | | | | | | |
| Major and clinically relevant non-major bleeding | 15 | 33.3 | 11 | 24.4 | 13 | 27.1 | 0.67 |
| Major bleedings | 0 | 0 | 2 | 4.4 | 0 | 0 | 0.21 |
| Fatal bleeding | 0 | 0 | 0 | 0 | 0 | 0 | |
| Clinically overt bleeding, haemoglobin (Hb) decrease >20 g/L | 0 | 0 | 0 | 0 | 0 | 0 | |
| Clinically overt bleeding, transfusion | 0 | 0 | 1 | 2.2 | 0 | 0 | 0.65 |
| Critical bleeding | 0 | 0 | 1* | 2.2 | 0 | 0 | 0.65 |
| Bleeding warranting treatment cessation | 0 | 0 | 1* | 2.2 | 0 | 0 | 0.65 |
| Bleeding located at surgical site | 0 | 0 | 0 | 0 | 0 | 0 | |
| Clinically relevant non-major bleeding | 15 | 33.3 | 10 | 22.2 | 13 | 27.1 | 0.51 |
| Spontaneous skin haematoma >25 cm$^2$ | 1 | 2.2 | 0 | 0 | 0 | 0 | 0.65 |
| Wound haematoma >100 cm$^2$ | 14 | 31.1 | 9 | 20.0 | 12 | 25.0 | 0.49 |
| Spontaneous nose bleeding >5 min | 0 | 0 | 1 | 2.2 | 0 | 0 | 0.65 |
| Spontaneous rectal bleeding >1 spot | 0 | 0 | 0 | 0 | 0 | 0 | |
| Macroscopic haematuria >24 hours | 0 | 0 | 0 | 0 | 1 | 2.1 | 1.0 |
| Other bleeding events (not major) | 0 | 0 | 0 | 0 | 0 | 0 | |
| Last primary outcome (median days after surgery) (range) | 7 (1–21) | | 7 (2–26) | | 6 (2–16) | | 0.32 |
| Patients with primary outcome more than 14 days after surgery (%) | 1 | 2.2 | 1 | 2.2 | 1 | 2.1 | 1.0 |
| **Secondary outcome** | | | | | | | |
| VTE | 1* | 2.2 | 0 | 0 | 0 | 0 | 0.65 |
| Death due to VTE | 1* | | 0 | | 0 | | |
| DVT | 0 | | 0 | | 0 | | |
| PE | 1* | | 0 | | 0 | | |
| Hospital stay (median, days) (range) | 3 (2–9) | | 3 (2–10) | | 4 (2–28) | | 0.098 |
| Readmission within 3 months, n (%) | 2 (4.4) | | 4 (8.9) | | 3 (6.3) | | 0.77 |
| Patients with adverse reactions, n (%) | 9 (20) | | 2 (4.4) | | 5 (10.4) | | 0.072 |
| Patient compliant, n (%) | 27 of 44 (61.4)† | | 25 of 44 (56.8)† | | 25 (52.1) | | 0.66 |
| Days of missed dosages (median, days) (range) | 0 (0–7); n=42 | | 0 (0–23); n=40 | | 0 (0–4); n=40 | | 0.73 |

*Identical patient.
†One missing value. Due to the explorative nature of this pilot study, all p values are explorative rather than confirmative and are not corrected for multiple testing.
DVT, deep venous thrombosis; PE, pulmonary embolism; VTE, venous thromboembolism.

patient-reported outcomes as assessed with the KOOS was found. The hypothesis was that joint haemorrhages could impair subjective improvement after TKA as assessed by KOOS. Our pilot study shows that it is feasible to assess KOOS in a study evaluating bleeding complications of thromboprophylaxis. However, variability in KOOS is probably too high to detect differences in KOOS in a future multicentre clinical trial.

The results of this pilot study should be interpreted with caution due to some limitations. First, as a pilot study, the size of the study population is limited. Although our aim was to conduct a pilot study to design a multicentre study with an improved external validity compared with available randomised clinical trials, we had to exclude a large number of patients some of who will receive thromboprophylaxis with NOACs in regular clinical practice. Additionally, we had to exclude patients with a high risk of bleeding who may receive thromboprophylaxis with NOACs in regular clinical practice.

**Table 3** Knee Injury and Osteoarthritis Outcome Score (KOOS) for patients with dabigatran, nadroparin or rivaroxaban thromboprophylaxis after total knee arthroplasty. Difference between 6 weeks postsurgery and presurgery KOOS and proportion of patients with 10 points or more improvement in KOOS between 6 weeks postsurgery and presurgery

| | Dabigatran | Nadroparin | Rivaroxaban | P value |
|---|---|---|---|---|
| Pain | n=32 | n=35 | n=35 | |
| Difference between 6 weeks postsurgery and presurgery (mean±SD) | 25±26 | 19±22 | 22±23 | 0.51 |
| Patients with ≥10 points improvement postsurgery versus presurgery, n (%) | 22 (69) | 26 (74) | 27 (77) | 0.78 |
| Symptoms | n=32 | n=35 | n=35 | |
| Difference between 6 weeks postsurgery and presurgery (mean±SD) | 12±20 | 14±20 | 16±19 | 0.63 |
| Patients with ≥10 points improvement postsurgery versus presurgery, n (%) | 17 (53) | 20 (57) | 22 (63) | 0.72 |
| ADL | n=32 | n=35 | n=35 | |
| Difference between 6 weeks postsurgery and presurgery (mean±SD) | 25±20 | 23±21 | 21±17 | 0.75 |
| Patients with ≥10 points improvement postsurgery versus presurgery, n (%) | 23 (72) | 25 (71) | 27 (77) | 0.88 |
| Sport and recreation | n=29 | n=31 | n=29 | |
| Difference between 6 weeks postsurgery and presurgery (mean±SD) | 24±38 | 25±41 | 28±33 | 0.89 |
| Patients with ≥10 points improvement postsurgery versus presurgery, n (%) | 17 (59) | 21 (68) | 23 (79) | 0.24 |
| Quality of life | n=31 | n=35 | n=35 | |
| Difference between 6 weeks postsurgery and presurgery (mean±SD) | 26±27 | 18±20 | 22±21 | 0.39 |
| Patients with ≥10 points improvement postsurgery versus presurgery, n (%) | 24 (77) | 23 (66) | 23 (66) | 0.55 |

Due to the explorative nature of this pilot study, all p values are explorativerather than confirmative and are not corrected for multiple testing.
ADL, activities of daily living function.

## CONCLUSION

A multicentre clinical trial may be feasible based on the recruitment speed and satisfaction with the trial protocol. No differences in major and clinically relevant non-major bleeding events were found in a pilot study of the head-to-head comparison of dabigatran, nadroparin and rivaroxaban as 42-day thromboprophylaxis in TKA. After the first 14-day use of dabigatran, nadroparin or rivaroxaban, bleeding risks were low and not different. This finding implies that conducting a multicentre trial will require substantial investment to achieve an adequate number of evaluated patients. Furthermore, due to the high variability in the KOOS, this outcome score may not be suitable to detect functional loss due to bleeding in a future clinical trial.

## Author affiliations
[1]Department of Clinical Pharmacy and Toxicology, Martini Hospital, Groningen, The Netherlands
[2]Department of Clinical Pharmacy, Ommelander Hospital Groningen, Winschoten, The Netherlands
[3]Department of Orthopaedic Surgery, Martini Hospital, Groningen, The Netherlands
[4]Department of Epidemiology, University of Groningen, University Medical Center Groningen, Groningen, The Netherlands
[5]Department of Health Sciences, University of Groningen, University Medical Center Groningen, Groningen, The Netherlands

**Acknowledgements** The adjudication committee (Mrs A. van der Aart, hospital pharmacist; Mrs A. van der Velden, internist–haematologist/oncologist; and Dr R.W. Brouwer, orthopaedic surgeon) is thanked for the careful assessment of bleeding events. Dr Brouwer joined the writing committee after adjudication of bleeding events. Mrs Roya Sorghabi is thanked for clinical trial management and data acquisition. Nurses and doctors of the orthopaedic surgery department are thanked for screening patients for bleeding events.

**Contributors** LvdV, MvH, CLEG-B and JJAMvR conceived the idea of the study and were responsible for the design of the study. NV, LvdV and MvH performed the statistical analysis for the study. The initial draft of the manuscript was prepared by LvdV, MS, MvH, JJAMvR, RWB and NJGMV and then circulated repeatedly among all authors for critical revision. LvdV and MS were responsible for the acquisition of the data, and MvH and NJGMV contributed to the interpretation of the results. LvdV was the coordinator of the study and lead writer. All authors helped plan the study, evolve analysis plans and critically revise successive drafts of the manuscript.

**Funding** The authors have not declared a specific grant for this research from any funding agency in the public, commercial or not-for-profit sectors.

**Competing interests** MvH reports grants from Bayer and personal fees from Boehringer Ingelheim during the conduct of the study. Other authors have nothing to report.

**Patient consent for publication** Not required.

**Ethics approval** This study was approved by the medical ethical committee 'Regionale Toetsingscommissie Patiëntgebonden Onderzoek' (RTPO) in Leeuwarden, the Netherlands (NL37609.099.12/RTPO 785).

**Provenance and peer review** Not commissioned; externally peer reviewed.

**Data availability statement** Data are available upon reasonable request. Data can made be available upon request to the corresponding author.

**ORCID iD**
Marinus van Hulst http://orcid.org/0000-0003-3216-7246

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
