## [Reviewer comments · BMJ Open]

ARTICLE DETAILS

TITLE (PROVISIONAL)	Bleeding complications of thromboprophylaxis with dabigatran, nadroparin or rivaroxaban for six weeks after total knee arthroplasty surgery – a randomised pilot study
AUTHORS	van der Veen, Lucia; Segers, Marijn; van Raay, Jos; Gerritsma-Bleeker, Carina; Brouwer, Reinoud; Veeger, Nic; van Hulst, Marinus

VERSION 1 – REVIEW

REVIEWER	Yoon Kong Loke University of East Anglia
REVIEW RETURNED	22-May-2020

GENERAL COMMENTS	Thank you for giving me the opportunity to comment on this submission. I have a few suggestions: 1) Please list in the Limitations if there were any post-hoc analyses, or modifications to the protocol during the conduct of the trial. 2) There is too much statistical significance testing in this manuscript, with so many p-values being reported in the Results section. This is a pilot trial that was never powered to reliably test any hypotheses. You can simply report results as being similar amongst the groups. You run into multiplicity of testing errors from doing so many tests. 3) To help other researchers, your pilot study has generated some effect estimates that can be used to model or predict sample sizes in a larger definitive trial. You should try to present some idea of the sample size needed for a non-inferiority trial based on bleeding endpoint, or on KOOS. This will help to justify your conclusions that it may be expensive or difficult to successfully run an RCT. 4) Can you comment, when comparing bleeding rates with other trials, whether you felt your method of capturing or detecting such events was more comprehensive? This may explain higher rates, in addition to longer duration and wider case definition.
---

REVIEWER	RODRIGO RIBEIRO PINHO RODARTE INSTITUTO NACIONAL DE ORTOPEDIA E TRAUMATOLOGIA
REVIEW RETURNED	18-Oct-2020

GENERAL COMMENTS	The research shows some problems that must be correct 1 - References are too old, there are a lot of recent articles 2 - The written English is not acceptable for publication 3 - The statistics used is not appropriate and must be reviewed
--

VERSION 1 – AUTHOR RESPONSE

Reviewer: 1

Comments to the Author

Thank you for giving me the opportunity to comment on this submission. I have a few suggestions:

1) Please list in the Limitations if there were any post-hoc analyses, or modifications to the protocol during the conduct of the trial.

Dear reviewer, thank you for the suggestion. As there were no post-hoc analyses, or modifications to the protocol, we did not specify these in the Limitations section.

2) There is too much statistical significance testing in this manuscript, with so many p-values being reported in the Results section. This is a pilot trial that was never powered to reliably test any hypotheses. You can simply report results as being similar amongst the groups. You run into multiplicity of testing errors from doing so many tests.

Thank you, we fully agree that too many p-values are reported in the Results section. We now only mention the p-values only for the primary endpoint. Furthermore, in the legend of Table 2 we included an additional disclaimer that p-values should be seen as explorative rather than confirmative and are not corrected for multiple testing.

3) To help other researchers, your pilot study has generated some effect estimates that can be used to model or predict sample sizes in a larger definitive trial. You should try to present some idea of the sample size needed for a non-inferiority trial based on bleeding endpoint, or on KOOS. This will help to justify your conclusions that it may be expensive or difficult to successfully run an RCT.

Thank for the suggestion. Although including a sample size calculation in a pilot study is debated, we have now included the expected sample size in the results section, please see page 10, line 14.

4) Can you comment, when comparing bleeding rates with other trials, whether you felt your method of capturing or detecting such events was more comprehensive? This may explain higher rates, in addition to longer duration and wider case definition.

Thank you for this question. Adhering strictly to the EMA definition of bleeding, and thus including wound haematomas > 100 cm², mainly explains the higher bleeding rates found in our study. We also added Anderson et al. Aspirin or Rivaroxaban for VTE Prophylaxis after Hip or Knee Arthroplasty N Engl J Med 2018; 378:699-707 to highlight this difference.

Reviewer: 2

Comments to the Author

The research shows some problems that must be correct

1 - References are too old, there are a lot of recent articles

Thank you for the suggestion. We have now included the most important recent paper “Anderson et al. Aspirin or Rivaroxaban for VTE Prophylaxis after Hip or Knee Arthroplasty N Engl J Med 2018; 378:699-707” in our discussions.

2 - The written English is not acceptable for publication

Thank you for your comment. We have reviewed the paper for any improvements that can be made in the written English.

3 - The statistics used is not appropriate and must be reviewed

Thank you for your comment. Unfortunately, your comment is rather unspecific. The statistical methods were reviewed by an independent statistician are deemed appropriate.